# Mixability made efficient:
# Fast online multiclass logistic regression

**Rémi Jézéquel**
INRIA - Département d'Informatique de l'École
Normale Supérieure, PSL Research University
Paris, France
remi.jezequel@inria.fr

**Pierre Gaillard**
Univ. Grenoble Alpes, Inria,
CNRS, Grenoble INP, LJK
Grenoble, France
pierre.gaillard@inria.fr

**Alessandro Rudi**
INRIA - Département d'Informatique de l'École
Normale Supérieure, PSL Research University
Paris, France
alessandro.rudi@inria.fr

## Abstract

Mixability has been shown to be a powerful tool to obtain algorithms with optimal regret. However, the resulting methods often suffer from high computational complexity which has reduced their practical applicability. For example, in the case of multiclass logistic regression, the aggregating forecaster (Foster et al. (2018)) achieves a regret of $O(\log(Bn))$ whereas Online Newton Step achieves $O(e^B \log(n))$ obtaining a double exponential gain in $B$ (a bound on the norm of comparative functions). However, this high statistical performance is at the price of a prohibitive computational complexity $O(n^{37})$.

In this paper, we use quadratic surrogates to make aggregating forecasters more efficient. We show that the resulting algorithm has still high statistical performance for a large class of losses. In particular, we derive an algorithm for multi-class logistic regression with a regret bounded by $O(B \log(n))$ and a computational complexity of only $O(n^4)$.

## 1   Introduction

In online learning, a learner sequentially interacts with an environment and tries to learn based on data observed on the fly (Cesa-Bianchi and Lugosi (2006), Hazan et al. (2016)). More formally, at each iteration $t \geq 1$, the learner receives an input $x_t$ in some space $\mathcal{X}$; makes a prediction $\widehat{y}_t$ in a decision domain $\widehat{\mathcal{Y}}$ and the environment reveals the output $y_t \in \mathcal{Y}$. The inputs $x_t$ and the outputs $y_t$ are sequentially chosen by the environment and can be arbitrary. No stochastic assumption (except boundedness) on the data sequence $(x_t, y_t)_{1 \leq t \leq n}$ is made. The accuracy of a prediction $\widehat{y}_t \in \widehat{\mathcal{Y}}$ at instant $t \geq 1$ for the outcome $y_t \in \mathcal{Y}$ is measured through a loss function $\ell : \widehat{\mathcal{Y}} \times \mathcal{Y} \to \mathbb{R}$. The learner aims at minimizing his cumulative regret

$$R_n(f) = \sum_{t=1}^{n} \ell(\widehat{y}_t, y_t) - \sum_{t=1}^{n} \ell(f(x_t), y_t), \tag{1}$$

with respect to any function $f \in \mathcal{F}$ in a reference class. All along this paper, we will consider parametric class of functions $f_\theta$ indexed by $\theta \in \Theta \subset \mathbb{R}^d$. We will also assume convexity of the loss according to index $\theta$, which we denote $\ell_{x,y} : \theta \mapsto \ell(f_\theta(x), y)$ for any $(x, y) \in \mathcal{X} \times \mathcal{Y}$. In this

35th Conference on Neural Information Processing Systems (NeurIPS 2021).

| Algorithm | OGD | ONS | Foster et al. (2018) | **GAF** (Ours) |
|---|---|---|---|---|
| Regret | $B\sqrt{n}$ | $dKe^B \log(n)$ | $dK \log(Bn)$ | $dK(B^2 + B\log(n))$ |
| Total complexity | $ndK$ | $nd^3K^3$ | $B^6 n^{25}(Bn + dK)^{12}$ | $nK^3 d^2 + K^2 n^4$ |

Table 1: Regret bounds and computational complexities (in $O(\cdot)$) of relevant algorithms for logistic regression with $K$ classes.

general context, many algorithms have been designed based on different assumptions and obtaining different trade-offs for regret. We review below the most relevant ones for our purpose.

Assuming only convexity of the loss $\ell$, a well-known strategy for the learner is Online Gradient Descent (Zinkevich (2003)) with an optimal regret of order $O(\sqrt{n})$. However, if the loss is $\alpha$-mixable, the learner may achieve the faster rate $O(\frac{1}{\alpha} \log(n))$ by using an aggregating forecaster (see Vovk (2001), Van Erven et al. (2015)). Yet, such an algorithm is not constructive in general and when it is, the computational complexity is often very high (a notable exception is the least-squares setting). To reduce it, a stronger assumption has been introduced by Hazan et al. (2007) with $\eta$-exp concavity. Under the latter hypothesis, Online Newton Step (ONS) has a regret bounded by $O(\frac{1}{\eta} \log(n))$ and a computational complexity of $O(n)$. However this efficiency comes at the price of deteriorating statistical performance. Indeed, $\eta$-exp concavity implies $\alpha$-mixability for $\eta \geq \alpha$ and in some cases, the gap between $\eta$ and $\alpha$ can be very large.

The most spectacular case of this phenomenon occurs for logistic regression. In this setting, the loss is defined as

$$\ell(\widehat{y}_t, y_t) = -\log(\sigma(\widehat{y}_t)_{y_t}) \qquad \text{where} \qquad \sigma(z)_i = \frac{e^{z_i}}{\sum_j e^{z_j}}$$

and the regret is computed with respect to linear functions $\mathcal{F} = \{x \mapsto Wx, W \in \Theta\}$, where $\Theta = \mathcal{B}(\mathbb{R}^{K \times d}, B)$ is the Frobenius bounded ball of radius $B > 0$. On this subset, the logistic loss is $\eta$-exp concave *only* for $\eta \leq e^{-B}$. The regret of Online Newton Step can thus be of order $O(e^B \log(n))$. On the other hand, as remarked by Kakade and Ng (2008) (binary case) and Foster et al. (2018), the logistic loss is 1-mixable and an aggregating algorithm may achieve $O(\log(nB))$. Nevertheless, the algorithm relies on Monte Carlo methods and must sample from log-concave probability distributions which is extremely computationally expensive (see Table 1). Therefore, in this framework, the exp-concavity assumption leads to an efficient algorithm with low dependence on $B$ while mixability yields an inefficient algorithm with much better statistical performance.

Between these two extremes, other algorithms and trade-offs have been analyzed. First, it has been shown that in some situations, Follow The Regularized Leader (FTRL) can achieve a fast rate without an exponential constant (Bach et al. (2010), Marteau-Ferey et al. (2019), Ostrovskii et al. (2021)). However, several additional assumptions are necessary to achieve these rates. Unfortunately, they are essentially unavoidable. Indeed, Hazan et al. (2014) showed a polynomial lower bound for the proper algorithm (*i.e.,* with linear prediction function) in the regime $B = \log(n)$. This prevents algorithms like ONS and Follow The Regularized Leader from reaching logarithmic regret without an exponential constant in $B$. This result motivates the search for improper algorithms with better computational complexity than Foster et al. (2018).

For binary logistic regression, two efficient improper algorithms have been proposed in the literature. First, in the i.i.d. framework, Mourtada and Gaïffas (2019) have proposed the Sample Minmax Predictor (SMP). It achieves an excess risk of order $O((B^2 + d)/n)$ with computational complexity equivalent to Follow The Regularized leader. In the online framework, Jézéquel et al. (2020) proposed AIOLI, which is based on quadratic approximations of the logistic loss as well as virtual labels to regularize. The regret is upper-bounded $O(dB \log(n))$ and the computational complexity is $O(n(d^2 + \log(n)))$.

These previous works left open the question of achieving the same type of performance in a setting other than binary logistic regression. In particular, for multi-class logistic regression, no other algorithm than Foster et al. (2018) is known to achieve logarithmic regret without an exponential constant.

**Contributions** We introduce in Section 3 a new generic online learning algorithm, that we call Gaussian Aggregating Forecaster (GAF) . GAF achieves logarithmic regret with a small multiplicative constant for a large class of convex loss functions (see Theorem 1). The latter includes several popular loss functions such as squared loss, binary and multi-class logistic loss. Our assumptions on the loss functions are slightly stronger than $\alpha$-mixability but generally weaker than assumptions widely used in the statistical framework such as generalized self-concordance (see Bach et al. (2010)).

In the particular but significant setting of multi-class logistic regression, GAF has a regret bounded by $O(dKB^2 + dKB \log(n))$ and a total computational complexity of $O(nK^3d^2 + K^2n^4)$, thus significantly improving on the $O(B^6 n^{25}(Bn + dK)^{12})$ complexity of Foster et al. (2018), which was the best known to date for algorithms without exponential dependence on $B$. Table 1 summarizes the regrets and computational cost obtained by the relevant algorithms in this framework. It is worth pointing out that, by using standard online-to-batch conversion Helmbold and Warmuth (1995), this paper also provides new results on the excess risk in the statistical i.i.d. Indeed, even in this extensively used framework, the only existing improper algorithms without exponential constant in $B$ so far are GAF and Foster et al. (2018). Jézéquel et al. (2020) and Mourtada and Gaïffas (2019) are in fact restricted to binary outputs only.

Our new algorithm is inspired by the aggregating forecaster of Vovk (2001) that we apply to quadratic approximations of the loss, in order to make it more efficient. The high-level idea is that these quadratic approximations replace complicated log-concave distributions with Gaussian distributions, from which it is much easier to sample and thereby drastically reducing the complexity of the algorithm. We believe that this approach can be applied to much broader contexts than those analyzed here.

## 2 Setting

We recall here the setting and introduce the main notations and assumptions that will be used throughout the paper.

**Setting and notation** Our framework is formalized as a sequential game between a learner and an environment. At each forecasting instance $t \geq 1$, the learner is given an input $x_t \in \mathcal{X}$; forms a prediction $\widehat{y}_t \in \widehat{\mathcal{Y}} \subseteq \mathbb{R}^K$ (possibly based on the current input $x_t$ and on the past information $x_1, y_1, \ldots, x_{t-1}, y_{t-1}$). Note that the prediction space $\widehat{\mathcal{Y}} \subseteq \mathbb{R}^K$ may be uni-dimensional ($K = 1$) in some settings (least-square regression) or multi-dimensional ($K \geq 1$) in some cases (e.g., vector-valued regression, multi-class classification). Then, the environment chooses $y_t \in \mathcal{Y}$; reveals it to the learner which incurs the loss $\ell(\widehat{y}_t, y_t)$. Finally, the performance of the learner is assessed by the regret

$$R_n(\theta) = \sum_{t=1}^{n} \ell(\widehat{y}_t, y_t) - \sum_{t=1}^{n} \ell_{x_t, y_t}(\theta),$$

with respect to all $\theta \in \Theta \subset \mathbb{R}^d$. Here, $\ell_{x,y}(\theta) = \ell(f_\theta(x), y)$ where $\mathcal{F} = \{f_\theta : \mathcal{X} \to \widehat{\mathcal{Y}}\}$ is the reference class of functions. For simplicity of notation, we also define $\ell_t(\theta) = \ell_{x_t, y_t}(\theta)$ and $L_t(\theta) = \sum_{s=1}^{t} \ell_s(\theta) + \lambda \|\theta\|^2$ for all $t \geq 1$ and $\lambda > 0$.

In all specific examples considered in this work, the learner will be compared to linear functions $f_\theta : x \mapsto \theta^\top \Phi(x)$, where $\theta \in \Theta = \mathcal{B}(\mathbb{R}^d, B)$ and $\Phi$ is a function from $\mathcal{X}$ to $\mathbb{R}^{d \times K}$ such that for any $x \in \mathcal{X}$ and $i \in [K]$, $\|\Phi(x)_{.,i}\| \leq R$. We introduce and discuss below our main assumptions on the losses.

**Assumptions** We assume that, for all $(x, y) \in \mathcal{X} \times \mathcal{Y}$, the losses $\ell_{x,y}$ are convex, $\mathcal{C}^2$ and satisfy the following assumptions.

(A1) The loss function $\ell$ is $\alpha$-mixable. In other words, for all (Gaussian) probability distributions $\pi$ over $\mathbb{R}^d$ and input $x \in \mathcal{X}$, there exists $\widehat{y} \in \widehat{\mathcal{Y}}$ such that for all $y \in \mathcal{Y}$,

$$\ell(\widehat{y}, y) \leq -\frac{1}{\alpha} \log \left( \mathbb{E}_{\theta \sim \pi} e^{-\alpha \ell_{x,y}(\theta)} \right).$$

(A2) There exists $\zeta > 0$ such that, for all $(x, y) \in \mathcal{X} \times \mathcal{Y}$ and $\theta_1, \theta_2 \in \mathbb{R}^d$,

$$\ell_{x,y}(\theta_1) \leq \ell_{x,y}(\theta_2) + \nabla \ell_{x,y}(\theta_2)^\top (\theta_1 - \theta_2) + e^{\zeta \|\theta_1 - \theta_2\|^2} \|\theta_1 - \theta_2\|_{\nabla^2 \ell_{x,y}(\theta_2)}^2.$$

(A3) There exists $\beta > 0$ such that, for all $(x, y) \in \mathcal{X} \times \mathcal{Y}$ and $(\theta_1, \theta_2) \in \Theta \times \mathbb{R}^d$,

$$\ell_{x,y}(\theta_1) \geq \ell_{x,y}(\theta_2) + \nabla \ell_{x,y}(\theta_2)^\top (\theta_1 - \theta_2) + \frac{\beta}{2}(\theta_1 - \theta_2)^\top \nabla^2 \ell_{x,y}(\theta_2)(\theta_1 - \theta_2).$$

(A4) $\ell$ is $\gamma$-smooth *i.e.,* for all $x \in \mathcal{X}, y \in \mathcal{Y}, \theta \in \mathbb{R}^d, \nabla^2 \ell_{x,y}(\theta) \leq \gamma I$.

In general, these assumptions are rather weak and related to standard assumptions of (online) convex optimization. (A1) follows from instance from exp-concavity. As we illustrate in Section 4, all are satisfied for broadly used loss functions such as squared loss, binary and multi-class logistic loss. We provide more details on these assumptions below.

Mixability (A1) is a standard assumption in online learning to achieve fast rates for the regret (Van Erven et al., 2015), which was introduced by Vovk (2001). It is a weaker condition than exp-concavity because (A1) holds with $\widehat{y} = \mathbb{E}_{\theta \sim \pi}[\theta]$ for $\eta$-exp concave loss functions when $\eta \geq \alpha$.

Assumption (A2) basically prevents the Hessian from changing too quickly on $\theta$. This condition is new but not very restrictive because, as we prove in Lemma 6, it is weaker than other widely accepted assumptions such as generalized self-concordance (Bach et al., 2010).

The lower bound (A3) is the most original assumption and is crucial to make aggregating forecaster efficient. This bound is close to the one needed in the analyses of ONS, which can be derived from $\eta$-exp-concavity. However, the key difference is that (A3) uses the Hessian in the quadratic term and not the outer product of the gradient. This makes the right-hand-side closer to the third Taylor series approximation of the loss and allows much larger values for $\beta$. For example, as we will show in the section 4.2, this change allows us to remove the exponential constant for the logistic regression setting. In this case, (A3) is verified with $\beta \simeq B^{-1}$, while $\eta$-exp-concavity would require $\eta \simeq e^{-B}$.

## 3 Algorithm and regret bound

GAF is a new sequential forecasting rule inspired by the aggregating forecaster of Vovk (2001). It may be implemented if the loss function $\ell$ is mixable (A1) and ensures fast performance guarantees on the regret under Assumptions (A1)-(A4) as shown in Theorem 1. GAF requires the following hyper-parameters: a regularization parameter $\lambda > 0$ and $\alpha, \beta > 0$ such that the loss $\ell$ satisfies assumptions (A1)-(A4).

**Algorithm (GAF)** At each time step, the prediction $\widehat{y}_t$ is formed by GAF by following a two steps procedure. A first estimator $\theta_t \in \mathbb{R}^d$ is computed by solving

$$\theta_t = \underset{\theta \in \mathbb{R}^d}{\operatorname{argmin}} \left\{ \sum_{s=1}^{t-2} \tilde{\ell}_s(\theta) + \ell_{t-1}(\theta) + \lambda \|\theta\|_2^2 \right\}. \tag{2}$$

Basically, $\theta_t$ follows a regularized leader where the losses $\ell_s$ for $s \leq t-2$ are substituted with quadratic approximations $\tilde{\ell}_s$. Then, GAF computes the quadratic approximation of $\ell_{t-1}$ at point $\theta_t$. For any $\theta \in \mathbb{R}^d$, it defines

$$\tilde{\ell}_{t-1}(\theta) = \ell_{t-1}(\theta_t) + \nabla \ell_{t-1}(\theta_t)^\top (\theta - \theta_t) + \frac{\beta}{2}(\theta - \theta_t)^\top \nabla^2 \ell_{t-1}(\theta_t)(\theta - \theta_t). \tag{3}$$

Finally, GAF predicts $\widehat{y}_t \in \widehat{\mathcal{Y}}$ such that for all $y \in \mathcal{Y}$,

$$\ell(\widehat{y}_t, y) \leq -\frac{1}{\alpha} \log\left( \mathbb{E}_{\theta \sim \tilde{P}_{t-1}} e^{-\alpha \ell_{x_t, y}(\theta)} \right), \qquad \text{where} \qquad \tilde{P}_{t-1}(\theta) = \frac{e^{-\alpha \tilde{L}_{t-1}(\theta)}}{\int_{\mathbb{R}^d} e^{-\alpha \tilde{L}_{t-1}(\theta)} d\theta}, \tag{4}$$

with $\tilde{L}_{t-1}(\theta) = \sum_{s=1}^{t-1} \tilde{\ell}_s(\theta) + \lambda \|\theta\|^2$. Such a prediction $\widehat{y}_t$ exists as soon as Assumption (A1) is true. In Section 4, we present some specific cases where $\widehat{y}_t$ can be computed in a closed form.

We now state our main theoretical result, which is an upper bound on the regret suffered by GAF.

**Theorem 1.** *Let $d, n \geq 1$, $B > 0$, and $\Theta \subset \mathcal{B}(\mathbb{R}^d, B)$. Let $(x_1, y_1), ..., (x_n, y_n) \in \mathcal{X} \times \mathcal{Y}$ be an arbitrary sequence of observations and $\ell$ a loss function that verifies Assumptions (A1)-(A4) with $\alpha, \zeta, \beta, \gamma > 0$. GAF (4), run with regularization parameter $\lambda \geq \max\{4, d\}\zeta\alpha^{-1}$, satisfies the following upper-bound on the regret*

$$R_n(\theta) \leq \lambda \|\theta\|^2 + \frac{d}{\alpha} \left[ \frac{1}{2} + \frac{2\sqrt{3}}{\beta} \right] \log \left( 1 + \frac{n\beta\gamma}{2\lambda} \right), \qquad \forall \theta \in \Theta.$$

*In particular, the choice $\lambda = \max\{4, d\}\zeta\alpha^{-1}$ yields*

$$R_n(\theta) \leq \max\{4, d\} \frac{\zeta B^2}{\alpha} + \frac{d}{\alpha} \left[ \frac{1}{2} + \frac{2\sqrt{3}}{\beta} \right] \log \left( 1 + \frac{n\alpha\beta\gamma}{d\zeta} \right), \qquad \forall \theta \in \Theta.$$

Theorem 1 states that GAF has a logarithmic regret in the number of samples with a multiplicative constant proportional to $\alpha^{-1}(1 + \beta^{-1})$. Recalling that $\alpha$ is the mixability parameter, $\alpha^{-1}$ is the optimal multiplicative constant that can be obtained using an (computationally expensive) aggregating forecaster. The constant $\beta$ is the curvature parameter in the quadratic lower-bound assumption (A3). Having $\beta^{-1}$ as a multiplicative constant can be seen as similar to the regret bound of ONS. However, it is important to note the crucial difference that, unlike ONS, it is the Hessian that is used in the quadratic substitutes of Assumption (A3) and not the gradient outer product. Thus, in some cases, it is possible to have a much larger value for $\beta$, than would be attainable using the exp-concavity. For example, for multiclass logistic regression, the loss is 1-mixable (Proposition 1, Foster et al., 2018) but only $e^{-B}$-exp concave. We prove in Lemma 4 that this loss verifies Assumption (A3) with parameter $\beta \simeq (\log(K) + BR)^{-1}$. Therefore, GAF achieves, in the logistic case, a regret upper bounded by $O(dB^2R^2 + d(\log(K) + BR)\log(n))$. More details on this specific case are given in Section 4.2.

We provide below a sketch of the proof of Theorem 1 that highlights the key steps in the proof. The full proof is available in Appendix B.

*Sketch of proof.* The proof starts from the definition (4) of the prediction $\widehat{y}_t$ formed by the algorithm. It satisfies the mixability assumption (A1) for all $y \in \mathcal{Y}$

$$\ell(\widehat{y}_t, y) \leq -\frac{1}{\alpha} \log \left( \mathbb{E}_{\theta \sim \tilde{P}_{t-1}(\theta)} e^{-\alpha \ell_t(\theta)} \right), \quad \text{where} \quad \tilde{P}_{t-1}(\theta) = \frac{e^{-\alpha \tilde{L}_{t-1}(\theta)}}{\int_{\mathbb{R}^d} e^{-\alpha \tilde{L}_{t-1}(\theta)} d\theta}.$$

Next, we decompose the term on the right side of the inequality into two terms: an approximation error that will be small for well-chosen parameters; and a term, which telescopes and is standard for aggregating forecaster's analysis (Vovk, 2001), except that here it is written with quadratic surrogate losses. We have,

$$\ell(\widehat{y}_t, y_t) \leq -\frac{1}{\alpha} \log \left( \frac{\int_{\mathbb{R}^d} e^{-\alpha(\ell_t(\theta) - \tilde{\ell}_t(\theta)) - \alpha \tilde{L}_t(\theta)} d\theta}{\int_{\mathbb{R}^d} e^{-\alpha \tilde{L}_{t-1}(\theta)} d\theta} \right)$$

$$= \underbrace{-\frac{1}{\alpha} \log \left( \mathbb{E}_{\theta \sim \tilde{P}_t} e^{-\alpha[\ell_t(\theta) - \tilde{\ell}_t(\theta)]} \right)}_{\Omega_t} + \underbrace{\frac{1}{\alpha} \log \left( \frac{\int_{\mathbb{R}^d} e^{-\alpha \tilde{L}_{t-1}(\theta)} d\theta}{\int_{\mathbb{R}^d} e^{-\alpha \tilde{L}_t(\theta)} d\theta} \right)}_{\Psi_t}.$$

Now, the core idea is that since $\tilde{L}_t$ is quadratic, the integrals inside $\Psi_t$ are Gaussian, which allows closed form formulas for the analysis (and fast computational time). Using that $\nabla \tilde{\ell}_t(\theta_{t+1}) = \nabla \ell_t(\theta_{t+1})$ and the expression of Gaussian integrals, it is possible to show that

$$\Psi_t = \tilde{L}_t(\theta_{t+1}) - \tilde{L}_{t-1}(\theta_t) + \frac{1}{2\alpha} \log \left( \frac{|A_t|}{|A_{t-1}|} \right),$$

where $A_t$ is the Hessian of $\tilde{L}_t/2$. Therefore, summing over $t = 1, \ldots, n$ and using $\tilde{L}_0(\theta_1) = 0$, it yields

$$\sum_{t=1}^{n} \ell(\widehat{y}_t, y_t) - \tilde{L}_n(\theta_{n+1}) \leq \sum_{t=1}^{n} \Omega_t + \frac{1}{2\alpha} \log \left( \frac{|A_n|}{|A_0|} \right).$$

But, from Assumption (A3) together with the definition (2) of $\theta_{t+1}$, for any $\theta \in \Theta$, $L_t(\theta) \geq \tilde{L}_t(\theta) \geq \tilde{L}_t(\theta_{t+1})$. Thus, the regret can be bounded for any $\theta \in \Theta$ by

$$R_n(\theta) = \sum_{t=1}^{n} \ell(\widehat{y}_t, y_t) - L_n(\theta) \leq \lambda \|\theta\|^2 + \sum_{t=1}^{n} \Omega_t + \frac{1}{2\alpha} \log\left(\frac{|A_n|}{|\lambda I|}\right). \tag{5}$$

Finally, Assumption (A2) allows to bound the approximation error $\Omega_t$ by a telescopic term close to the usual one but with $\frac{1}{\alpha\beta}$ as multiplicative constant instead of $\frac{1}{\alpha}$,

$$\Omega_t \leq \frac{C}{\alpha\beta} \log\left(\frac{|A_t|}{|A_{t-1}|}\right).$$

The proof is concluded by applying Assumption (A4). $\qquad\square$

## 4 Specific settings: squared loss and multi-class logistic

In this section we show how our general framework cover several interesting settings. In particular, we prove that our Assumption (A1)-(A4) are satisfied and provide concrete implementations of GAF in those contexts.

### 4.1 Squared loss

For linear regression with squared loss, we recover classical results by Vovk (2001). This framework is defined, as a special case of our generic setting of Section 2, by setting: input domain $\mathcal{X} = \mathcal{B}(\mathbb{R}^d, R)$, output domain $\mathcal{Y} = [-Y, Y]$ with $Y > 0$, decision domain $\widehat{\mathcal{Y}} = \mathbb{R}$, loss function $\ell(\widehat{y}_t, y_t) = (\widehat{y}_t - y_t)^2$ and input feature $\Phi(x) = x$. Now we can verify that all assumptions are true:

(A1) By Lemma 2 and Lemma 3 of Vovk (2001), the squared loss is mixable with parameter $\alpha = 1/(2Y^2)$.

(A2) The inequality is true for any $\zeta > 0$ as $e^x > 1/2$ for all $x > 0$. We will take $\zeta = (8Y^2 dB^2)^{-1}$.

(A3) It holds with $\beta = 1$ because the Taylor expansion of order 2 of a quadratic function is an equality.

(A4) It is true with $\gamma = R^2$ since $\nabla^2 \ell_{x,y}(\theta) = 2xx^\top \leq 2R^2 I$.

In particular, in this particular case the quadratic approximations are exact $\tilde{\ell}_t = \ell_t$ for all $t \geq 1$. Therefore, GAF reduces to the classical non-linear Ridge regression of Vovk (2001) and Azoury and Warmuth (2001) that predicts

$$\widehat{y}_t = \widehat{\theta}_t^\top x_t \quad \text{with} \quad \widehat{\theta}_t = \operatorname{argmin}_{\theta \in \mathbb{R}^d} \left\{ L_{t-1}(\theta) + \theta^\top x_t + \lambda \|\theta\|^2 \right\}.$$

A direct consequence of Theorem 1, by substituting the constants derived above, yields

$$R_n(\theta) \leq 1 + 8Y^2 d \log\left(1 + 16nB^2 R^2\right).$$

This result essentially recovers the existing regret bound for the non-linear Ridge regression (see e.g., Theorem 11.8 of Cesa-Bianchi and Lugosi (2006)).

### 4.2 Logistic regression

The most interesting application of our general framework is logistic regression. In this section, we show the validity of our assumptions and the concrete implementation of GAF in this context. We present the theoretical results obtained and discuss the computational complexity.

The goal of logistic regression is to form a $K$-dimensional prediction $\widehat{y} \in \widehat{\mathcal{Y}} = \mathbb{R}^K$ of a categorical label $y \in \mathcal{Y} = \{1, ..., K\}$ from the observation of an input label $x \in \mathcal{X} = \mathcal{B}(\mathbb{R}^{d'}, R)$, $d' \geq 1$. The performance of $\widehat{y}$ is measured by the logistic loss defined by

$$\ell(\widehat{y}, y) = -\log(\sigma(\widehat{y})_y) \quad \text{where} \quad \sigma(z)_i = \frac{e^{z_i}}{\sum_j e^{z_j}}.$$

Defining the input feature $\Phi(x) \in \mathbb{R}^{d \times K}$ with $d = d'K$ and the linear predictions respectively as

$$\Phi(x)_{i,j} = \begin{cases} x_{i-d'j} & \text{if } d'j \leq i < d'(j+1) \\ 0 & \text{otherwise} \end{cases} \quad \text{and} \quad f_\theta(x) = \theta^\top \Phi(x), \quad \text{for } \theta \in B(\mathbb{R}^d, B),$$

one can check that our setting recovers the standard multi-class logistic regression setting (see Foster et al. (2018) for an equivalent convention). We check below Assumptions (A1)–(A4).

Parameters $\lambda, \mu, \beta, \varepsilon, \delta > 0, m \geq 1$
initialize $A_0 = \lambda I, b_0 = 0, \theta_1 = 0$
**for** $t = 1, ..., n$ **do**
$\quad$ receive $x_t \in \mathbb{R}^d$
$\quad$ sample independently $\omega_1, ..., \omega_m \sim \mathcal{N}(\theta_t^\top \Phi(x_t), \Phi(x_t)^\top A_{t-1}^{-1} \Phi(x_t))$
$\quad$ predict $\tilde{y}_t = \sigma^+(\text{smooth}_\mu(\frac{1}{m} \sum_{i=1}^m \sigma(\omega_i)))$ where $\sigma(z)_i = \frac{e^{z_i}}{\sum_j e^{z_j}}, \sigma^+(z)_i = \log(z_i)$ and
$\quad\quad$ $\text{smooth}_\mu(z) := (1 - \mu)z + \mu\mathbf{1}/K$.
$\quad$ receive $y_t \in \{1, ..., K\}$
$\quad$ update $\quad \theta_{t+1} = \arg\min_{\theta \in \mathbb{R}^{Kd}} b_{t-1}^\top \theta + \theta^\top A_{t-1}\theta + \ell_{x_t, y_t}(\theta)$
$\quad\quad\quad\quad\quad b_t = b_{t-1} + \nabla \ell_{x_t, y_t}(\theta_{t+1}) - \beta \nabla^2 \ell_{x_t, y_t}(\theta_{t+1})\theta_t$
$\quad\quad\quad\quad\quad A_t = A_{t-1} + \frac{\beta}{2} \nabla^2 \ell_{x_t, y_t}(\theta_{t+1})$
**end**

$$\text{\textbf{Algorithm 1}: Efficient-GAF for } K\text{-class logistic regression}$$

(A1) It holds with $\alpha = 1$, since by Proposition 1 of Foster et al. (2018), the logistic loss is 1-mixable. Indeed, given a distribution $\pi$ on $\mathbb{R}^K$, the choice $\widehat{y}_\pi = \sigma^+(\mathbb{E}_{\widehat{y} \sim \pi} \sigma(\widehat{y}))$, where $\sigma^+(z)_k := \log(z_k)$, satisfies

$$\mathbb{E}_{\widehat{y} \sim \pi} \exp(-\ell(\widehat{y}, y)) = \mathbb{E}_{\widehat{y} \sim \pi} \sigma(\widehat{y})_y = \sigma(\widehat{y}_\pi)_y = \exp(-\ell(\widehat{y}_\pi, y)),$$

for any $y \in [K]$.
(A2) It is true for $\zeta = 4R^2$ by Lemma 6 in Appendix C.
(A3) It holds with $\beta = (\log(K)/2 + BR + 1)^{-1}$ by Lemma 4 in Appendix C applied with the choices $a = \theta_1^\top \Phi(x)$ and $b = \theta_2^\top \Phi(x) \in [-BR, BR]^K$.
(A4) It is valid with $\gamma = R^2$. Indeed, $\nabla^2 \ell_{x,y}(\theta) = \Phi(x)(\text{diag}(p) - pp^\top)\Phi(x)^\top$ where $p = \sigma(\theta^\top \Phi(x))$ which can be bounded by $\nabla^2 \ell_{x,y}(\theta) \leq \Phi(x)\Phi(x)^\top \leq R^2 I$.

In particular, the proof of the mixability assumption (A1) provides us a concrete expression for Equation (4) of GAF by setting

$$\widehat{y}_t = \sigma^+(\mathbb{E}_{\theta \sim \tilde{P}_{t-1}(\theta)}(\sigma(\theta^\top x_t))). \tag{6}$$

Substituting the above parameters into Theorem 1 yields the following corollary.

**Corollary 2.** *Let $R, B > 0$ and $d', n, K \geq 1$. Let $(x_1, y_1), ..., (x_n, y_n) \in \mathcal{X} \times \mathcal{Y}$ be an arbitrary sequence of observations and $\ell(\widehat{y}, y) = -\log(\sigma(\widehat{y})_y)$. GAF, run with $\lambda = 32d'KR^2$, $\beta = (\log(K)/2 + BR + 1)^{-1}$, and $\alpha = 1$, satisfies*

$$R_n(\theta) \lesssim d'KB^2R^2 + d'K[\log(K) + BR]\log\left(1 + \frac{n}{d'K(\log(K) + BR)}\right), \tag{7}$$

*for all $\theta \in \mathcal{B}(\mathbb{R}^d, B)$, where $\lesssim$ denotes an approximate inequality which is up to universal multiplicative and additive constants.*

**Computationally efficient approximation** Yet, a key difficulty remains to compute exactly $\widehat{y}_t$ in Equation (6): the calculation of the expectation. In general, there is no closed-form expression and an approximation algorithm, such as Monte Carlo sampling, must be used. We provide now a fully implementable approximated version of GAF, which satisfies the same regret guarantees up to negligible additive constants. It is described in Algorithm 1.

First, to ensure that our approximation of the expectation produces forecasts close to those of the exact algorithm, we must smooth the function $\sigma^+$ in Equation (6). Following the idea of Foster et al. (2018), we thus define for some $\mu \in [0, \frac{1}{2}]$, the smoothing operator $\text{smooth}_\mu : \Delta_K \to \Delta_K$ by

$$\text{smooth}_\mu(p) = (1 - \mu)p + \mu\mathbf{1}/K, \qquad \forall p \in \Delta_k.$$

Now, we can show that $\sigma^+(\text{smooth}_\mu(\cdot))$ is $\mu$-Lispchitz. Denoting the smoothed approximation of $\widehat{y}_t$ by $\bar{y}_t = \sigma^+(\text{smooth}_\mu(\mathbb{E}_{\theta \sim \tilde{P}_{t-1}}(\sigma(\theta x_t)))$, Lemma 16 of Foster et al. (2018) shows that using $\bar{y}_t$ instead of $\widehat{y}_t$ worsens the regret bound only by an additional constant $\mu n$; hence our choice $\mu = n^{-1}$. The last step of the approximated algorithm consists in using Monte Carlo sampling to approximate the expectation by a finite sum

$$\tilde{y}_t = \sigma^+\left(\text{smooth}_\mu\left(\frac{1}{m}\sum_{i=1}^m \sigma(\omega_i)\right)\right) \qquad \text{where} \quad \omega_i \overset{\text{i.i.d.}}{\sim} \mathcal{N}(\theta_t^\top \Phi(x_t), \Phi(x_t)^\top A_{t-1}^{-1}\Phi(x_t)).$$

Using Chernoff's inequality, we show below that $\tilde{y}_t$ concentrates well to $\bar{y}_t$ which entails the following guarantee.

**Proposition 3.** *Let $\delta > 0$. Efficient-GAF, run with parameters $\lambda = 32d'KR^2$, $\beta = (\log(K)/2 + BR + 1)^{-1}$, $m \geq 1$, and $\mu = (\log(n/\delta)/m)^{1/3}$ satisfies with probability $1 - \delta$ the regret bound*

$$R_n(\theta) \lesssim d'KB^2R^2 + d'K\left[\log(K) + BR\right]\log\left(1 + \frac{n}{d'K(\log(K) + BR)}\right) + n\left(\frac{K}{m}\log\left(\frac{n}{\delta}\right)\right)^{\frac{1}{3}}$$

*with a computational cost $O(nK^3d^2 + K^2nm)$. In particular, the choice $m = n^3\log(n/\delta)$, yields the regret bound (7) with probability $1 - \delta$ and a total computational time of order $O(nK^3d^2 + K^2n^4\log(n/\delta))$.*

*Proof.* We first prove the computational complexity upper-bound. At each iteration $t = 1, \ldots, n$, the algorithm performs the following computations:

(i) Update $A_t^{-1}$: since the rank of $A_t - A_{t-1}$ is $K$, the inverse $A_t^{-1}$ can be updated in $O(K^3d^2)$ operations.

(ii) Update $\theta_{t+1}$: by Theorem 4 of Jézéquel et al. (2020), this can be done in $O(\log(n) + K^2d^2)$.

(iii) Compute $\tilde{y}_t$: to do so, it must sample $m$ times from a Gaussian of dimension $K$. The cost is thus $O(mK^2)$.

Therefore, the overall time complexity of the algorithm is $O(nK^3d^2 + K^2nm)$.

We now prove the corresponding regret bound. First, we define

$$\tilde{p}_t := \text{smooth}_\mu\left(\frac{1}{m}\sum_{i=1}^m \sigma(\omega_i)\right) \qquad \text{and} \qquad \bar{p}_t := \mathbb{E}[\tilde{p}_t] = \text{smooth}_\mu\left(\mathbb{E}_{\theta \sim \tilde{P}_{t-1}}(\sigma(\theta^\top \Phi(x_t)))\right).$$

With this notation, $\tilde{y}_t = \sigma^+(\tilde{p}_t)$ and $\bar{y}_t = \sigma^+(\bar{p}_t)$. Using that $\sigma(\sigma^+(p)) = p$ for any $p \in \Delta_K$ yields for all $y_t \in [K]$

$$\ell(\tilde{y}_t, y_t) = -\log\left(\sigma(\tilde{y}_t)_{y_t}\right) = -\log\left(\sigma(\sigma^+(\tilde{p}_t))_{y_t}\right) = -\log(\tilde{p}_{t,y_t}),$$

where $\tilde{p}_{t,i}$ denotes the $i$-th component of $\tilde{p}_t$. Similarly, $\ell(\bar{y}_t, y_t) = -\log(\bar{p}_{t,y_t})$. Therefore, fixing some $\varepsilon > 0$, we have

$$\mathbb{P}\left[\ell(\tilde{y}_t, y_t) - \ell(\bar{y}_t, y_t) > \varepsilon\right] = \mathbb{P}\left[-\log\left(\frac{\tilde{p}_{t,y_t}}{\bar{p}_{t,y_t}}\right) > \varepsilon\right] = \mathbb{P}\left[\tilde{p}_{t,y_t} - \bar{p}_{t,y_t} < (e^{-\varepsilon} - 1)\bar{p}_{t,y_t}\right].$$

Using that $e^{-\varepsilon} - 1 \leq 1 - 2\varepsilon$ for $\varepsilon \in (0, \frac{1}{2})$ together with the multiplicative Chernoff's bound (see e.g., Theorem 1.10.5 of Doerr (2020)), entails for all $0 < \varepsilon < \frac{1}{2}$,

$$\mathbb{P}\left[\ell(\tilde{y}_t, y_t) - \ell(\bar{y}_t, y_t) > \varepsilon\right] \leq \mathbb{P}\left[\tilde{p}_{t,y_t} < (1 - 2\varepsilon)\bar{p}_{t,y_t}\right] \leq e^{-2\varepsilon^2 m\bar{p}_{t,y_t}} \leq e^{-\varepsilon^2 m\mu/K}, \qquad (8)$$

where the last inequality is because $\text{smooth}_\mu(p)_i \geq \mu/K$ for all $i \in [K]$. Taking $\varepsilon = \sqrt{\frac{K}{\mu m}\log(n/\delta)}$, we have

$$\mathbb{P}\left[\ell(\tilde{y}_t, y_t) - \ell(\bar{y}_t, y_t) > \sqrt{\frac{K}{\mu m}\log\left(\frac{n}{\delta}\right)}\right] \leq \frac{\delta}{n}. \qquad (9)$$

Furthermore, denoting $\hat{p}_t = \mathbb{E}_{\theta \sim \tilde{P}_{t-1}}(\sigma(\theta^\top \Phi(x_t)))$ so that $\bar{p}_t = \text{smooth}_\mu(\hat{p}_t)$ and following Lemma 16 of Foster et al. (2018), we get

$$\ell(\bar{y}_t, y_t) - \ell(\hat{y}_t, y_t) = \log\left(\frac{\hat{p}_{t,y_t}}{\bar{p}_{t,y_t}}\right) = \log\left(\frac{\hat{p}_{t,y_t}}{\text{smooth}_\mu(\hat{p}_t)_{y_t}}\right) = \log\left(\frac{\hat{p}_{t,y_t}}{(1 - \mu)\hat{p}_{t,y_t} + \frac{\mu}{K}}\right)$$

$$= \log\left(\frac{1}{1 - \mu\left(1 - \frac{1}{K\hat{p}_{t,y_t}}\right)}\right) \leq 2\mu\left(1 - \frac{1}{K\hat{p}_{t,y_t}}\right) \leq 2\mu. \qquad (10)$$

Combining (9) and (10), and using a union bound yields with probability at least $1 - \delta$

$$\sum_{t=1}^n \ell(\tilde{y}_t, y_t) - \ell(\hat{y}_t, y_t) \leq n\sqrt{\frac{K}{\mu m}\log\left(\frac{n}{\delta}\right)} + \mu n.$$

Optimizing $\mu = (\log(n/\delta)K/m)^{1/3}$ and using Corollary 3 concludes the proof. $\qquad \square$

Using Gaussian distribution improves significantly the computation time with respect to log-concave distribution as considered by Foster et al. (2018). Indeed, it allows to get exact and efficient samples from $\tilde{P}_{t-1}$ which leads to a computation time of order $O(n^4)$. On the contrary, since it is not possible to draw an exact sample from a log-concave distribution, Foster et al. (2018) must resort to expensive random walks to approximately sample from it. Using the method from Bubeck et al. (2018), as suggested by Foster et al. (2018), leads to a $O(B^6 n^{24} (Bn + dK)^{12})$ computation time per iteration (see their Example 3 with the choices $\varepsilon = n^{-2}$ and $\mu = n^{-1}$).

Note that our analysis shows a clean trade-off between the computational time and the regret bound. One could set a smaller value for $m$ at the price of a larger regret. In particular, it is worth pointing out that our analysis consider the worst case scenario. In the experiments that we considered, it seems that much less samples are sufficient to reach a good accuracy. This is partly because the lower-bound $\bar{p}_{t,y_t} \geq \mu \approx 1/n$ in Inequality (8) may be very coarse for many $t$. If the $\bar{p}_{t,y_t}$ were of order $\Omega(1)$, one could choose $m = O(n^2)$ instead of $m = O(n^3)$. Another possible explanation comes from the fact that, if $A_t$ is large enough, the variance of the samples is small and the convergence is much faster. In our experiments, the choice of $m = 100$ already provides a good approximation. We leave to future work the possibility to improve the complexity in favorable scenarios.

**Comparison with Jézéquel et al. (2020)**  In the binary case, a more efficient algorithm called AIOLI has been introduced in Jézéquel et al. (2020). The latter achieves a similar regret bound as GAF, while having a computational complexity of only $O(n(d^2 + \log(n)))$. This was possible because AIOLI does not rely on Monte Carlo sampling at all and uses only convex optimization instead. Therefore, one may wonder if it is possible to extend directly AIOLI to the multiclass setting avoiding Monte Carlo methods used in this work. In fact, we first tried to analyse the regret of natural extensions of AIOLI but we found the following inherent difficulties. On the intuitive side, AIOLI was based on the observation that when $\theta_t$ is far from 0 (let say $\theta_t \gg 0$) either $y_t = 1$ and the curvature was advantageous or $y_t = -1$ and $\theta_t$ tends to the oracle $\theta_{t+1}$. This intuition is a bit lost in the multiclass setting as several oracles are possible if $y_t \neq 1$. On a more technical side, the analysis in Jézéquel et al. (2020) crucially relies on the relation $g_t^{-y_t} = -(1 + BR)\eta_t g_t$ (Equation 20) which seems to have no equivalent in the multiclass setting. Thus, it remains an open question if an extension of AIOLI to the multiclass setting is possible.

## 5 Experiments

Although GAF is primarily theoretically motivated in a worst-case analysis, here we study its performance on real data sets. We consider three datasets (vehicle, shuttle, and segmentation taken from LIBSVM Data [1]) and compare the performance of GAF with two well-used algorithms: Online Gradient Descent (OGD) (Zinkevich, 2003) and Online Newton Step (ONS) (Hazan et al., 2007). The algorithm of Foster et al. (2018) is not considered because of prohibitive computational complexity. Concerning the hyper-parameters, the values suggested by the theory are generally too conservative. We thus choose the best ones in a grid for each algorithm ($\lambda, \beta \in [0.01, 0.03, 0.1, 0.3, 1, 3, 10]$).

The averaged losses over time are reported in Figure 1. We can remark that the performance of GAF is similar to the one of ONS when the number of samples is high. However, the learning of GAF seems more stable than ONS which leads sometimes (vehicle and segment) to better performance when there are few samples. This is not surprising as aggregating forecasters are hedging against worst-case scenario. Those results show that GAF is not only a theoretical oriented algorithm but could also be used successfully in practice. However, we expect it to perform best in a hard adversarial regime (close to the one described in Hazan et al. (2014)).

## Conclusion and future work

We have shown how to leverage both mixability and quadratic approximations to design an algorithm GAF which achieves high statistical performance while being more efficient than existing algorithms. In particular, it achieves a new trade-off for multiclass logistic regression.

Some interesting questions are still remaining and left for future work. The linear dependence in $B$ although better than what can be achieved by any proper algorithm is still sub-optimal compared

---

[1] https://www.csie.ntu.edu.tw/~cjlin/libsvmtools/datasets/multiclass.html

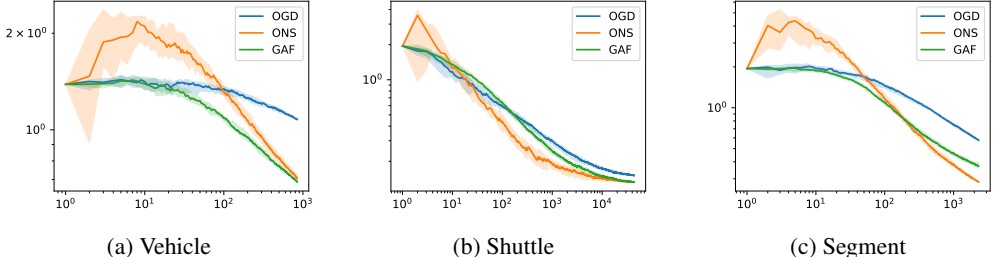

|  (a) Vehicle | (b) Shuttle | (c) Segment |

Figure 1: Averaged losses over time incurred by ONS, GAF, and OGD. The experiments were repeated 20 times and the empirical quantiles 0.25, 0.5, and 0.75 are reported.

to Foster et al. (2018) (logarithmic dependence). It may be possible to improve it by using other surrogates than quadratics. The essential point would be to prove an equivalent of Lemma 4 for those surrogates. The computational complexity of computing Equation (6) should also remain low. The computational complexity may also be improved. Like previous point, other surrogates could be used to make Equation (6) easier to compute. Finally, we believe that GAF should easily be extended to kernels. One would need to adapt the analysis to depend on the effective dimension (Jézéquel et al., 2019) of the RKHS instead of $d$. It would of interest to see how classical approximation algorithms, like Nystrom and random Fourier features, deteriorate statistical performance. Finally, we have only provided two examples of loss functions (multi-class logistic loss and squared loss). An intriguing question would be to see if the algorithm can be used to improve existing results for other losses such as Huber loss.

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
