# Supplementary material

## A   Notations and relevant equations

In this section, we give notations and useful identities which will be used in following proofs. At each forecasting instance $t \geq 1$, the learner is given an input $x_t \in \mathcal{X}$; forms a prediction $\widehat{y}_t \in \widehat{\mathcal{Y}}$. Then, the environment chooses $y_t \in \mathcal{Y}$; reveals it to the learner which incurs the loss $\ell(\widehat{y}_t, y_t)$. The regret is defined as

$$R_n(\theta) = \sum_{t=1}^{n} \ell(\widehat{y}_t, y_t) - \sum_{t=1}^{n} \ell_{x_t, y_t}(\theta),$$

with respect to all $\theta \in \Theta \subset \mathbb{R}^d$. Here, $\ell_{x,y}(\theta) = \ell(f_\theta(x), y)$ where $\mathcal{F} = \{f_\theta : \mathcal{X} \to \widehat{\mathcal{Y}}\}$ is the reference class of functions. We assume that, for all $(x, y) \in \mathcal{X} \times \mathcal{Y}$, the losses $\ell_{x,y}$ are convex, $\mathcal{C}^2$ and satisfies Assumptions (A1)-(A4) with parameters $\alpha, \zeta, \beta, \gamma > 0$. We also use the following notations:

- $\ell_t(\theta) = \ell_{x_t, y_t}(\theta)$
- $\theta_t = \operatorname{argmin}_{\theta \in \mathbb{R}^d} \left\{ \sum_{s=1}^{t-2} \tilde{\ell}_s(\theta) + \ell_{t-1}(\theta) + \lambda \|\theta\|_2^2 \right\}$
- $\tilde{\ell}_{t-1}(\theta) = \ell_{t-1}(\theta_t) + \nabla \ell_{t-1}(\theta_t)^\top (\theta - \theta_t) + \frac{\beta}{2}(\theta - \theta_t)^\top \nabla^2 \ell_{t-1}(\theta_t)(\theta - \theta_t)$
- $L_t(\theta) = \sum_{s=1}^{t} \ell_s(\theta) + \lambda \|\theta\|^2$ and $\tilde{L}_t(\theta) = \sum_{s=1}^{t} \tilde{\ell}_s(\theta) + \lambda \|\theta\|^2$
- $\tilde{P}_{t-1}(\theta) = \frac{e^{-\alpha \tilde{L}_{t-1}(\theta)}}{\int_{\mathbb{R}^d} e^{-\alpha \tilde{L}_{t-1}(\theta)} d\theta}$
- $A_t = \sum_{s=1}^{t} \frac{\beta}{2} \nabla^2 \ell_s(\theta_{s+1}) + \lambda I$ .

## B   Main proof

*Proof of Theorem 1.* Let $t \geq 1$. By definition, the prediction $\widehat{y}_t$ (see Equation (4)) satisfies the Mixability property (A1). Applied in $y = y_t$, it yields

$$\ell(\widehat{y}_t, y_t) \leq -\frac{1}{\alpha} \log \left( \mathbb{E}_{\theta \sim \tilde{P}_{t-1}(\theta)} e^{-\alpha \ell_t(\theta)} \right), \quad \text{where} \quad \tilde{P}_{t-1}(\theta) = \frac{e^{-\alpha \tilde{L}_{t-1}(\theta)}}{\int_{\mathbb{R}^d} e^{-\alpha \tilde{L}_{t-1}(\theta)} d\theta}$$

$$= -\frac{1}{\alpha} \log \left( \frac{\int_{\mathbb{R}^d} e^{-\alpha(\ell_t(\theta) - \tilde{\ell}_t(\theta)) - \alpha \tilde{L}_t(\theta)} d\theta}{\int_{\mathbb{R}^d} e^{-\alpha \tilde{L}_{t-1}(\theta)} d\theta} \right)$$

$$= -\frac{1}{\alpha} \log \left( \mathbb{E}_{\theta \sim \tilde{P}_t} e^{-\alpha[\ell_t(\theta) - \tilde{\ell}_t(\theta)]} \right) + \frac{1}{\alpha} \log \left( \frac{\int_{\mathbb{R}^d} e^{-\alpha \tilde{L}_{t-1}(\theta)} d\theta}{\int_{\mathbb{R}^d} e^{-\alpha \tilde{L}_t(\theta)} d\theta} \right) . \tag{11}$$

We recall that $\tilde{L}_t : \theta \mapsto \sum_{s=1}^{t} \tilde{\ell}_s(\theta) + \lambda I$ where for all $s \geq 1$

$$\tilde{\ell}_s(\theta) = \ell_s(\theta_{s+1}) + \nabla \ell_s(\theta_{s+1})^\top (\theta - \theta_{s+1}) + \frac{\beta}{2}(\theta - \theta_{s+1})^\top \nabla^2 \ell_s(\theta_{s+1})(\theta - \theta_{s+1}).$$

Thus, $\tilde{L}_t$ is a quadratic function with Hessian $2A_t$ where

$$A_t = \sum_{s=1}^{t} \frac{\beta}{2} \nabla^2 \ell_s(\theta_{s+1}) + \lambda I . \tag{12}$$

Moreover, by definition (2) of $\theta_{t+1}$ and since $0 = \nabla \tilde{L}_{t-1}(\theta_{t+1}) + \nabla \ell_t(\theta_{t+1}) = \nabla \tilde{L}_t(\theta_{t+1})$, we have

$$\theta_{t+1} = \operatorname*{argmin}_{\theta \in \mathbb{R}^d} \tilde{L}_t(\theta) . \tag{13}$$

Therefore,
$$\tilde{L}_t(\theta) = (\theta - \theta_{t+1})^\top A_t(\theta - \theta_{t+1}) + \tilde{L}_t(\theta_{t+1}),$$
and recognizing the integral of a multivariate Gaussian distribution $\tilde{P}_t \sim \mathcal{N}(\theta_{t+1}, \frac{1}{2\alpha}A_t^{-1})$, we get
$$\int_{\mathbb{R}^d} e^{-\alpha\tilde{L}_t(\theta)}d\theta = e^{-\alpha\tilde{L}_t(\theta_{t+1})} \int_{\mathbb{R}^d} e^{-\alpha(\theta-\theta_{t+1})^\top A_t(\theta-\theta_{t+1})}d\theta = \sqrt{(\pi/\alpha)^d|A_t^{-1}|}e^{-\alpha\tilde{L}_t(\theta_{t+1})},$$
which substituted into (11) gives
$$\ell(\widehat{y}_t, y_t) + \tilde{L}_{-1}(\theta_t) - \tilde{L}_t(\theta_{t+1}) \le \underbrace{-\frac{1}{\alpha}\log\left(\mathbb{E}_{\theta\sim\tilde{P}_t}e^{-\alpha(\ell_t(\theta)-\tilde{\ell}_t(\theta))}\right)}_{\Omega_t} + \frac{1}{2\alpha}\log\left(\frac{|A_t|}{|A_{t-1}|}\right).$$

Summing over $t$ and using $\tilde{L}_0(\theta_1) = 0$, the sum telescopes,
$$\sum_{t=1}^n \ell(\widehat{y}_t, y_t) - \tilde{L}_n(\theta_{n+1}) \le \sum_{t=1}^n \Omega_t + \frac{1}{2\alpha}\log\left(\frac{|A_n|}{|A_0|}\right).$$

Denote $L_n(\theta) = \sum_{t=1}^n \ell_t(\theta) + \lambda\|\theta\|^2$. By Assumption (A3) followed by (13), for all $\theta \in \mathbb{R}^d$
$$L_n(\theta) \overset{(A3)}{\ge} \tilde{L}_n(\theta) \overset{(13)}{\ge} \tilde{L}_n(\theta_{n+1}).$$

Therefore, plugging into the previous inequality, the regret can be bounded as
$$R_n(\theta) = \sum_{t=1}^n \ell(\widehat{y}_t, y_t) - L_n(\theta) + \lambda\|\theta\|_2^2 \le \lambda\|\theta\|_2^2 + \sum_{t=1}^n \Omega_t + \frac{1}{2\alpha}\log\left(\frac{|A_n|}{|\lambda I|}\right). \qquad (14)$$

Now it remains to bound the approximation terms $\Omega_t$. Using Jensen's inequality and the concavity of log, yields
$$\begin{aligned}
\Omega_t :&= -\frac{1}{\alpha}\log\left(\mathbb{E}_{\theta\sim\tilde{P}_t}e^{-\alpha(\ell_t(\theta)-\tilde{\ell}_t(\theta))}\right) \\
&\le \mathbb{E}_{\theta\sim\tilde{P}_t}\left[\ell_t(\theta) - \tilde{\ell}_t(\theta)\right] \\
&= \mathbb{E}_{\theta\sim\tilde{P}_t}\left[\ell_t(\theta) - \ell_t(\theta_{t+1}) - \nabla\ell_t(\theta_{t+1})^\top(\theta - \theta_{t+1}) - \frac{\beta}{2}(\theta-\theta_{t+1})^\top\nabla^2\ell_t(\theta_{t+1})(\theta-\theta_{t+1})\right] \\
&\le \mathbb{E}_{\theta\sim\tilde{P}_t}\left[\ell_t(\theta) - \ell_t(\theta_{t+1}) - \nabla\ell_t(\theta_{t+1})^\top(\theta - \theta_{t+1})\right].
\end{aligned}$$

By Assumption (A2) and Cauchy-Schwartz inequality,
$$\begin{aligned}
\Omega_t &\le \mathbb{E}_{\theta\sim\tilde{P}_t}\left[e^{\zeta\|\theta-\theta_{t+1}\|^2}\|\theta-\theta_{t+1}\|_{\nabla^2\ell_t(\theta_{t+1})}^2\right] \\
&\le \underbrace{\sqrt{\mathbb{E}_{\theta\sim\tilde{P}_t}e^{2\zeta\|\theta-\theta_{t+1}\|^2}}}_{\Omega_{t,1}}\underbrace{\sqrt{\mathbb{E}_{\theta\sim\tilde{P}_t}\|\theta-\theta_{t+1}\|_{\nabla^2\ell_t(\theta_{t+1})}^4}}_{\sqrt{\Omega_{t,2}}}. \qquad (15)
\end{aligned}$$

Now remarking that $\tilde{P}_t = \mathcal{N}(\theta_{t+1}, \frac{1}{2\alpha}A_t^{-1})$, let us bound $\Omega_{t,1}$ the term on the left of the product. There exists an orthonormal basis $e_1, \ldots, e_d$ in $\mathbb{R}^d$ such that $\theta - \theta_{t+1}$ follows the same distribution as
$$\sum_{i=1}^d \sqrt{\frac{1}{2\alpha}\lambda_i(A_t^{-1})}X_i e_i \qquad \text{where} \quad X_i \overset{\text{i.i.d.}}{\sim} \mathcal{N}(0,1), i = 1, \ldots, d,$$
and $\lambda_i(A_t^{-1})$ denotes the $i$-th largest eigenvalue of $A_t^{-1}$. Thus, since $\lambda_i(A_t^{-1}) \le \lambda^{-1}$,
$$\begin{aligned}
\Omega_{t,1} = \sqrt{\mathbb{E}_{\theta\sim\tilde{P}_t}\left[e^{2\zeta\|\theta-\theta_{t+1}\|^2}\right]} &\le \sqrt{\prod_{i=1}^d \mathbb{E}\left[e^{\frac{\zeta}{\alpha}\lambda_i(A_t^{-1})X_i^2}\right]} \le \sqrt{\prod_{i=1}^d \mathbb{E}\left[e^{\frac{\zeta}{\alpha\lambda}X_i^2}\right]} \\
&= \left(\mathbb{E}_{X\sim\chi^2}\left[e^{\frac{\zeta}{\alpha\lambda}X}\right]\right)^{d/2}
\end{aligned}$$

Then, because $\lambda \geq 4\zeta\alpha^{-1}$ by assumption and using that the moment-generating function of the $\chi^2$ distribution (with one degree of freedom) is $\mathbb{E}_{X \sim \chi^2}\big[\exp(tX)\big] = (1-2t)^{-1/2}$ for $t < 1/2$ and thus $\mathbb{E}_{X \sim \chi^2}[\exp(X/4)] = \sqrt{2}$, the term can be further upper-bounded as

$$
\begin{aligned}
\Omega_{t,1} \leq \mathbb{E}_{X \sim \chi^2}\Big[e^{\frac{\zeta}{\alpha\lambda}X}\Big]^{d/2} &\leq \mathbb{E}_{X \sim \chi^2}\Big[e^{\frac{1}{4}X}\Big]^{\frac{2d\zeta}{\lambda\alpha}} &\leftarrow \qquad \text{Jensen's inequality} \\
&\leq 2^{\frac{d\zeta}{\lambda\alpha}} \\
&\leq 2 &\leftarrow \qquad \text{since} \quad \lambda \geq d\zeta\alpha^{-1}. \qquad (16)
\end{aligned}
$$

We now upper-bound $\Omega_{t,2}$ in (15),

$$
\Omega_{t,2} := \mathbb{E}_{\theta \sim \tilde{P}_t}\|\theta - \theta_{t+1}\|^4_{\nabla^2\ell_t(\theta_{t+1})} = \mathbb{E}_{\theta \sim \mathcal{N}\left(0, \frac{1}{2\alpha}A_t^{-1}\right)}(\theta^\top \nabla^2\ell_t(\theta_{t+1})\theta)^2 = \mathbb{E}_{\theta \sim \mathcal{N}(0, \Sigma_t)}\|\theta\|^4
$$

where $\Sigma_t = \frac{1}{2\alpha}(\nabla^2\ell_t(\theta_{t+1}))^{1/2}A_t^{-1}(\nabla^2\ell_t(\theta_{t+1}))^{1/2}$. If we write $\lambda_i$ the $i$-th largest eigenvalue of $\Sigma_t$, there exists an orthonormal basis $e_1, \ldots, e_d$ such that

$$
\begin{aligned}
\Omega_{t,2} = \mathbb{E}_{\theta \sim \mathcal{N}(0, \Sigma_t)}\big[\|\theta\|^4\big] &= \mathbb{E}_{(X_i) \stackrel{iid}{\sim} \mathcal{N}(0,1)}\left[\left\|\sum_{i=1}^d \sqrt{\lambda_i} X_i e_i\right\|^4\right] \\
&= \mathbb{E}_{(X_i) \stackrel{iid}{\sim} \mathcal{N}(0,1)}\left[\left(\sum_{i=1}^d \lambda_i X_i^2\right)^2\right] = \sum_{i=1}^d\sum_{j=1}^d \lambda_i\lambda_j \mathbb{E}_{(X_i) \stackrel{iid}{\sim} \mathcal{N}(0,1)}\big[X_i^2 X_j^2\big].
\end{aligned}
$$

Then remarking that $\mathbb{E}_{(X_i) \stackrel{iid}{\sim} \mathcal{N}(0,1)}[X_i^2 X_j^2]$ equals to 3 if $i = j$ and 1 otherwise, we get the following upper-bound

$$
\begin{aligned}
\Omega_{t,2} \leq 3\sum_{i,j} \lambda_i\lambda_j = 3\left(\sum_{i=1}^d \lambda_i\right)^2 &= 3\,\mathrm{Tr}(\Sigma_t)^2 = 3\,\mathrm{Tr}\left(\frac{1}{2\alpha}A_t^{-1}\nabla^2\ell_t(\theta_{t+1})\right)^2 \\
&= \frac{3}{\alpha^2\beta^2}\,\mathrm{Tr}\left(A_t^{-1}\frac{\beta}{2}\nabla^2\ell_t(\theta_{t+1})\right)^2.
\end{aligned}
$$

Then, by Lemma 7,

$$
\Omega_{t,2} \leq \frac{3}{\alpha^2\beta^2}\log\left(\frac{|A_t|}{\left|A_t - \frac{\beta}{2}\nabla^2\ell_t(\theta_{t+1})\right|}\right)^2 = \frac{3}{\alpha^2\beta^2}\log\left(\frac{|A_t|}{|A_{t-1}|}\right)^2. \qquad (17)
$$

Then combining equations (15), (16) and (17), we have

$$
\Omega_t \leq \frac{2\sqrt{3}}{\alpha\beta}\log\left(\frac{|A_t|}{|A_{t-1}|}\right) \qquad (18)
$$

which, by summing over $t = 1, \ldots, n$, telescopes

$$
\sum_{t=1}^n \Omega_t \leq \frac{2\sqrt{3}}{\alpha\beta}\log\left(\frac{|A_n|}{|A_0|}\right).
$$

Combining this upper bound with equation (14) yields

$$
R_n(\theta) \leq \lambda\|\theta\|^2 + \frac{1}{\alpha}\left(\frac{1}{2} + \frac{2\sqrt{3}}{\beta}\right)\log\left(\frac{|A_n|}{|\lambda I|}\right), \qquad (19)
$$

which concludes the proof since by (12) and Assumption (A4)

$$
|A_n| \stackrel{(12)}{=} \left|\lambda I + \frac{\beta}{2}\sum_{t=1}^n \nabla^2\ell_t(\theta_{t+1})\right| \stackrel{(A4)}{\leq} \left|\left(\lambda + \frac{n\gamma\beta}{2}\right)I\right| = \left(1 + \frac{n\gamma\beta}{2\lambda}\right)^d |\lambda I|.
$$

$\square$

## C  Technical Lemmas

The following lemma shows that the logistic loss satisfies Assumption (A3) with $\beta = (\log(K)/2 + BR + 1)^{-1}$. Indeed, it suffices to apply it with $a = \theta_1^\top \phi(x)$, $b = \theta_2^\top \phi(x)$ and $C = BR$. Indeed, one can check that

$$
\begin{aligned}
\ell_y(a) &= \ell_{x,y}(\theta_1) \\
\ell_y(b) &= \ell_{x,y}(\theta_2) \\
\nabla \ell_y(b)^\top (a - b) &= \nabla \ell_{x,y}(\theta_2)^\top (\theta_1 - \theta_2) \\
(a - b)^\top \nabla^2 \ell_y(b)(a - b) &= (\theta_1 - \theta_2)^\top \nabla^2 \ell_{x,y}(\theta_2)(\theta_1 - \theta_2),
\end{aligned}
$$

where the terms on the left correspond to the notation of Lemma 4 and the terms on the right to Assumption (A3).

**Lemma 4.** *Let $C > 0$, $a \in [-C, C]^K$, $y \in [K]$, and $b \in \mathbb{R}^K$. Denote $\ell_y(a) = -\log\left(\frac{e^{a_y}}{\sum_j e^{a_j}}\right)$. Then,*

$$
\ell_y(a) \geq \ell_y(b) + \nabla \ell_y(b)^\top (a - b) + \frac{1}{\log(K) + 2(C + 1)} (a - b)^\top \nabla^2 \ell_y(b)(a - b).
$$

*Proof.* We start the proof by rephrasing our objective. Noting that $\ell_y(a) = -a^\top e_y + \log(\sum_{j=1}^K e^{a_j})$, one can subtract the linear part on both sides of the inequality. Thus, it suffices to prove the inequality for the function $f : a \mapsto \log(\sum_{j=1}^K e^{a_j})$. Defining

$$
\xi(a, b) = f(a) - f(b) - \nabla f(b)^\top (a - b) - \frac{\beta}{2}(a - b)^\top \nabla^2 f(b)(a - b), \tag{20}
$$

with $\beta = (\log(K)/2 + C + 1)^{-1}$, it is thus enough to prove that $\xi(a, b) \geq 0$ for all $a \in [-C, C]^K$ and $b \in \mathbb{R}^K$. But, because $\xi(a, a) = 0$, the latter is implied by

$$
\nabla_b \xi(a, b)^\top (b - a) \geq 0.
$$

Substituting the definition (20) of $\xi(a, b)$, this can be rewritten as

$$
(1 - \beta)(b - a)^\top \nabla^2 f(b)(b - a) - \frac{\beta}{2} \nabla^3 f(b)[b - a, b - a, b - a] \geq 0
$$

where for all $h \in \mathbb{R}^K$,

$$
\nabla^3 f(b)[h, h, h] = \sum_{i,j,k} (\nabla^3 f(b))_{i,j,k} h_i h_j h_k. \tag{21}
$$

Rearranging the terms gives the following condition

$$
\nabla^3 f(b)[b - a, b - a, b - a] \leq 2\left(\frac{1}{\beta} - 1\right) \nabla^2 f(b)[b - a, b - a]. \tag{22}
$$

where $\nabla^2 f(b)[h, h] = \sum_{i,j} (\nabla^2 f(b))_{i,j} h_i h_j$. Let $p \in \Delta^K$ defined as $p_i = \frac{e^{b_i}}{\sum_{j=1}^K e^{b_j}}$. The first two derivatives of $f$ satisfy

$$
\nabla f(b) = p, \quad \nabla^2 f(b) = \mathrm{diag}(p) - pp^\top.
$$

Then using that $\frac{\partial}{\partial b_j} p_i = \mathbf{1}[i = j]p_i - p_i p_j$ and chain rules of the derivative, the third derivative may be computed as follows

$$
\begin{aligned}
(\nabla^3 f(b))_{i,j,k} &= \frac{\partial}{\partial b_k}\left(\mathbf{1}[i = j]p_i - p_i p_j\right) \\
&= \mathbf{1}[i = j]\frac{\partial p_i}{\partial b_k} - \frac{\partial p_i}{\partial b_k}p_j - p_i \frac{\partial p_j}{\partial b_k} \\
&= \mathbf{1}[i = j](\mathbf{1}[i = k]p_i - p_i p_k) - (\mathbf{1}[i = k]p_i - p_i p_k)p_j - p_i(\mathbf{1}[j = k]p_j - p_j p_k) \\
&= \mathbf{1}[i = j = k]p_i - \mathbf{1}[i = j]p_i p_k - \mathbf{1}[i = k]p_i p_j - \mathbf{1}[j = k]p_i p_j + 2p_i p_j p_k. \tag{23}
\end{aligned}
$$

Let $X$ be a random variable which takes the values $b_i - a_i$ with probability $p_i$, for $i = 1, \ldots, K$. Now, note that

$$\mathbb{E}[X^3] = \sum_{i=1}^K p_i (b_i - a_i)^3 = \sum_{i,j,k} \mathbf{1}[i = j = k] p_i (b_i - a_i)(b_j - a_j)(b_k - a_k),$$

$$\mathbb{E}[X^2]\mathbb{E}[X] = \left(\sum_{i=1}^K p_i (b_i - a_i)^2\right)\left(\sum_{j=1}^K p_k (b_k - a_k)\right)$$

$$= \left(\sum_{i,j} p_i (b_i - a_i)(b_j - a_j)\mathbf{1}[i = j]\right)\left(\sum_{j=1}^K p_k (b_k - a_k)\right)$$

$$= \sum_{i,j,k} \mathbf{1}[i = j] p_i p_k (b_i - a_i)(b_j - a_j)(b_k - a_k),$$

$$\mathbb{E}[X]^3 = \sum_{i,j,k} p_i p_j p_k (b_i - a_i)(b_j - a_j)(b_k - a_k).$$

Therefore, summing Equation (23) over $i, j, k$ and recognizing the above values of $\mathbb{E}[X^3]$, $\mathbb{E}[X^2]\mathbb{E}[X]$ and $\mathbb{E}[X]^3$, the term on the left-hand-side of Inequality (22) can be rewritten as

$$\nabla^3 f(b)[b - a, b - a, b - a] \overset{(21)}{=} \sum_{i,j,k} (\nabla^3 f(b))_{i,j,k} (b_i - a_i)(b_j - a_j)(b_k - a_k)$$

$$\overset{(23)}{=} \mathbb{E}[X^3] - 3\mathbb{E}[X^2]\mathbb{E}[X] + 2\mathbb{E}[X]^3$$

$$= \mathbb{E}[(X - \mathbb{E}[X])^3]. \tag{24}$$

Similarly,
$$\nabla^2 \ell(b)[b - a, b - a] = \mathbb{E}[X^2] - \mathbb{E}[X]^2 = \mathbb{E}[(X - E[X])^2]. \tag{25}$$

Substituting (24) and (25) and replacing $\eta = (\log(K)/2 + C + 1)^{-1}$ into Inequality (22), the latter can be rewritten in the following way

$$\mathbb{E}[(X - \mathbb{E}[X])^3] \leq (2C + \log K)\mathbb{E}[(X - \mathbb{E}[X])^2]. \tag{26}$$

Recall that $X$ takes values $b_i - a_i$ with probability $p_i \propto e^{b_i}$ and that by assumption $\|a\|_\infty \leq C$. Almost surely, $X$ is upper-bounded as

$$X \leq \max_{1 \leq i \leq K} \{b_i - a_i\} \leq C + \max_{1 \leq i \leq K} b_i$$

and by Lemma 5,

$$\mathbb{E}[X] = \frac{\sum_{i=1}^K e^{b_i}(b_i - a_i)}{\sum_{j=1}^K e^{b_j}} \overset{(\|a\|_\infty \leq C)}{\geq} -C + \frac{\sum_{i=1}^K e^{b_i} b_i}{\sum_{j=1}^K e^{b_j}} \overset{\text{(Lem. 5)}}{\geq} -C - \log K + \max_{1 \leq i \leq K} b_i.$$

Hence, almost surely
$$X - \mathbb{E}[X] \leq 2C + \log K,$$

which implies Inequality (26) and thus conclude the proof. $\qquad \square$

**Lemma 5.** *For all $b \in \mathbb{R}^K$,*

$$\frac{\sum_{i=1}^K e^{b_i} b_i}{\sum_{j=1}^K e^{b_j}} \geq \max_{1 \leq i \leq K} b_i - \log(K).$$

*Proof.* Let $p \in \Delta_K$ defined as $p_i = \frac{e^{b_i}}{\sum_{j=1}^K e^{b_j}}$. We can write the left term as

$$\sum_{i=1}^K p_i b_i = -\sum_{i=1}^K p_i \log(e^{-b_i}).$$

By concavity of the logarithm, it follows from Jensen inequality that

$$\sum_{i=1}^{K} p_i b_i \geq -\log\left(\frac{K}{\sum_{i=1}^{K} e^{b_i}}\right) = \log\left(\sum_{i=1}^{K} e^{b_i}\right) - \log(K) \geq \max_{1 \leq i \leq K} b_i - \log(K).$$

□

The following lemma shows that the logistic loss satisfies Assumption (A2). The proof follows from generalized self-concordance.

**Lemma 6.** *The logistic loss $\ell$ verifies for all $y \in [K]$, $x \in \mathbb{R}^d$ and $\theta_1, \theta_2 \in \mathbb{R}^d$,*

$$\ell_{x,y}(\theta_1) - \ell_{x,y}(\theta_2) - \nabla\ell_{x,y}(\theta_2)(\theta_1 - \theta_2) \leq e^{4R^2\|\theta_1-\theta_2\|^2}\|\theta_1 - \theta_2\|^2_{\nabla^2\ell_{x,y}(\theta_2)}.$$

*Proof.* By example 2 of Marteau-Ferey et al. (2019), the logistic loss is generalized self-concordant with coefficient $2R$. By equation (30) of proposition 4 of the same paper, using $\lambda = 0$ and $\mu = \delta_{(x,y)}$ we have

$$\ell_{x,y}(\theta_1) - \ell_{x,y}(\theta_2) - \nabla\ell_{x,y}(\theta_2)(\theta_1 - \theta_2) \leq \psi(\|\theta_1 - \theta_2\|)\|\theta_1 - \theta_2\|^2_{\nabla^2\ell_{x,y}(\theta_2)}$$

with $\psi(t) = (e^t - 1 - t)/t^2$. Using that $\psi(t) \leq e^{t^2}$ for $t \geq 0$ concludes the proof. □

The following lemma is a classical technical result of linear algebra.

**Lemma 7.** *Let $d \in \mathbb{N}$ and $A, B \in \mathbb{S}_+(\mathbb{R}^d)$ such that $A > B$, then*

$$\mathrm{Tr}(A^{-1}B) \leq \log\left(\frac{|A|}{|A - B|}\right).$$

*Proof.* Using the concavity of $X \mapsto \log|X|$, we have

$$\log\left(\frac{|A|}{|A - B|}\right) = \log|A(A - B)^{-1}| \geq \mathrm{Tr}(I - (A - B)A^{-1}) = \mathrm{Tr}(A^{-1}B).$$

□