# OpenReview forum: "Mixability made efficient: Fast online multiclass logistic regression"
_NeurIPS.cc/2021/Conference — NeurIPS 2021 Spotlight_

### Official Review · Reviewer_uznP · 2021-07-06

**Rating:** 6
**Confidence:** 2

**Summary:**

With online learning for multiclass logistic regression, its regret has a trade-off against the computational cost. For example, ONS can be computed in $O(n)$ time ($n$: number of instances) while the regret is $O(e^B \log n)$: exponentially increased against the radius of regret computation $B$. [Foster 2018] can provide the regret $O(\log(Bn))$ but requires $O(n^{37})$ time.
The proposed method can provide another trade-off between them, and achieved the regret $O(B^2 + B\log n)$ with the computational cost of $O(n^4)$. This is much faster than [Foster 2018] with acceptable computational cost.

**Limitations And Societal Impact:**

The paper seems to state adequately the impact and the limitation. It provided an essential theoretical improvement, while it is not always reflected for real data by an experiment.

**Main Review:**

It seems valuable that the paper found a new trade-off between the regret and the computational cost, that can improve conventional methods. Compared to OGD and ONS, the proposed method GAF improved the regret at the expense of increase of the computational cost. Compared to [Foster 2018], GAF greatly improved the computational cost at the expense of small increase of the regret. In addition, the paper experimentally showed regrets compared to OGD and ONS; it showed that GAF can provide smaller regret for many cases.

For this result, the reviewer considers that it is more useful to discuss the real-life usage. For example, how about experimenting GAF with various $m$? In case the computational cost of $O(n^4)$ is unacceptable, we may reduce $m$ to reduce the cost (at the expense of the regret). In such cases, we may be also interested in how the regret becomes large.

Questions and comments:

-   Overall:
    -   What is essentially difficult if we consider the multiclass logistic regression instead of the binary logistic regression? As far as I read Table 1, the difficulty is not found by $K$ between 2 and 3. Is it a difficulty in the algorithm?
-   Section 3, first paragraph:
    -   The paper states that we have three hyperparameters $\lambda$, $\alpha$ and $\beta$. However, it seems that $\lambda$ and $\beta$ are tuning hyperparameters while $\alpha$ is a hyperparameter automatically determined from $\ell$. It is good to clarify the difference.
-   Section 3, Expression (4), definition of $\tilde{P}_{t-1}(\theta)$:
    -   Doesn't $\theta$ in the denominator of the right hand depend on $\theta$ in the left hand? If so, consider using a different variable name.
-   Section 4.1, (A4):
    -   It seems that the definition of operator $\leq$ for matrices is not provided. Is it defined by the positive definiteness?
-   Section 5:
    -   Since it is an online setting (i.e., we cannot see all data values beforehand), how to choose the hyperparameters seems to be a difficult problem and thus it should be clarified. For example, it choosed hyperparameters by first $n^\prime$ instances, or it updated hyperparameters whenever retrieving $n^\prime$ instances.
    -   According to the source code, it seems that the hyperparameters are chosen beforehand. If they are chosen by whole data, then it does not reflect the online setting and therefore experimental results may not be appropriate.
-   Section 5:
    -   Does the following experimental setups match: those GAF did not work well experimentally, and those GAF is not theoretically advantageous compared to OGD and ONS?

**Time Spent Reviewing:**

2

---

> ### Author Response · Authors · 2021-08-10
> **Response to reviewer uznP**
>
> We thank the reviewer for the useful comments and constructive considerations. We are glad that the reviewer appreciates the value of the new trade-off between the regret and the computational cost, improving conventional methods. We address the reviewer’s comments below and hope our argument will make the reviewer better evaluate our paper:
>
> - Overall:
>   - We thank the reviewer for the important question. The main difficulty is that previous methods have been designed specifically for the binary setting and generalizing them was not easy. To achieve the same kind of results in multiclass setting, we had to design an algorithm substantially different from the one of JGR and based on different principles. Indeed, we tried hard for a few months to find natural extensions of JGR, before finding the following inherent difficulties that make such a road hardly practicable. On the intuitive side, Jézéquel et al. was based on the observation that when $\theta_t$ is far from 0 (let say $\theta_t >> 0$) either $y_t = 1$ and the curvature was advantageous or $y_t = -1$ and $\hat \theta_t$ tends to the oracle $\theta_{t+1}$. This intuition is a bit lost in multiclass setting as several oracles are possible if $y_t \neq 1$. On a more technical side, if we extend Jézéquel et al. in a natural way we pay $g_t^\top A_t^{-1} \sum_{y \neq y_t} g_t^{y}$. In the binary case $\sum_{y \neq y_t} g_t^{y} = g_t^{-y_t} = -(1 + BR) \eta_t g_t$ (equation 20) which is helpful in the analysis. Sadly, it seems that there is no such relation in the multiclass setting which blocks the analysis.
>
>
> - Section 3, Expression (4), definition of $\tilde{P}_{t-1}(\theta)$:
>   - Thanks for spotting this out, we will improve the notation here.
> - Section 4.1, (A4):
>   - Yes, the operator $\le$ here is indeed the partial ordering associated to the cone of positive semidefinite matrices. We will clarify it.
> - Section 5:
>   - Choosing good hyper-parameters is a general problem in online learning. Here, we opted for the easiest solution to choose the best hyper-parameters a posteriori. We agree it is not realistic but this protocol is at least fair to all algorithms. As our code will be available, a more ambitious empirical study may test it with more complex protocols.
> - Section 5:
>   - If I understand the question correctly, in general it is not possible to understand a priori which algorithm will be the best on a given learning task, unless we have some a priori information on the structure of the learning problem. However, in a practical scenario such information is rarely available.

---

> > ### Comment · Reviewer_uznP · 2021-08-20
> > **Thanks for responses**
> >
> > I would like to thank authors for responses.
> >
> > -   For "Overall":
> >     -   I have understood. The forms of the optimization problems are largely different between the binary logistic regression and the multiclass logistic regression. As a result, a method for the binary logistic regression cannot be necessarily applicable for multiclass logistic regression.
> > -   For "Section 5" first point:
> >     -   I understand that the hyper-parameter selection for online algorithms is difficult, however, from a viewpoint of real-life usage, the results by just choosing hyper-parameters a posteori are not so informative. For example, how about displaying all results for all examined hyper-parameters? (It may be difficult to draw as a figure ...) Or, if there is a standard algorithm of hyper-parameter selection for online algorithms, the results with the algorithm can be informative.

---

### Official Review · Reviewer_TSZ4 · 2021-07-09

**Rating:** 7
**Confidence:** 4

**Summary:**

This paper considers the online convex optimization setting with mixable losses. The exponential weights algorithm is known to have constant regret with mixable losses, but it is also known to have prohibitively high running time for certain mixable losses. A mixable loss function of interest is the logistic loss used in logistic regression, for which faster algorithms such as Online Newton Step (ONS) suffer a double exponentially larger regret bound compared to exponential weights.

As a solution the authors propose a mix between ONS and exponential weights called GAF. To update, the algorithm computes a quadratic surrogate loss akin to the surrogate loss used in ONS, but uses the actual hessian rather than an approximation to the hessian to compute the surrogate loss. Another interesting trick that the authors use is that the point around which the quadratic surrogate loss is defined: it is the minimizer of the $l_2$ regularization plus the loss of the previous round and the surrogate losses of all rounds preceding the previous round.  The predictions of the algorithms are the same predictions that would be used for exponential weights. While for arbitrary priors this would pose a computational problem, the algorithm uses a gaussian prior. As is observed in prior work, with a gaussian prior and quadratic losses the posterior of exponential weights is also a gaussian. This allows the authors to devise more efficient predictions.



**Limitations And Societal Impact:**

The authors adequately addressed the limitations and potential negative societal impact of their work.

**Main Review:**

The problem considered in this paper is an important problem and the proposed solution is elegant. While the additional assumptions over the mixability assumption appear to be tailored to the logistic loss, I think that the ideas behind the algorithm can be used for other problems as well.

The paper is well written. In particular, the ideas behind the algorithm and its analysis are well explained.

Another natural approach to the problem would be to extend the algorithm of Jézéquil et. al. (2020) to $K >2$, especially since it appears to be significantly faster in $K = 2$. I suppose that the authors have at least considered this approach and if so I would like to have an idea on the difficulties in extending their approach.

As a final comment, $\beta$ is used for both exp-concavity and in (A3). While it is stated that these are different at a glance one might think they are the same, so I recommend changing the notation.


minor comments
- I think there is a typo in the definition of $\tilde{L}_t$ in line 406. Should it not be $+\lambda ||\theta||_2^2$?


**Time Spent Reviewing:**

4.5

---

> ### Author Response · Authors · 2021-08-10
> **Response to reviewer TSZ4**
>
> We thank the reviewer for the constructive comments and the typo l.406. We are glad that the reviewer appreciated the elegance of the proposed solution. We will now answer the main question on the difficulty to generalize Jézéquel et al.
>
> In our first attempt to the problem, we tried to extend Jézéquel et al. to the multiclass case, but without success encountering the following difficulties. On the intuitive side, Jézéquel et al. was based on the observation that when $\theta_t$ is far from 0 (let say $\theta_t >> 0$) either $y_t = 1$ and the curvature was advantageous or $y_t = -1$ and $\hat \theta_t$ tends to the oracle $\theta_{t+1}$. This intuition is a bit lost in multiclass setting as several oracles are possible if $y_t \neq 1$. On a more technical side, if we extend Jézéquel et al. in a natural way, we pay $g_t^\top A_t^{-1} \sum_{y \neq y_t} g_t^{y}$. In the binary case $\sum_{y \neq y_t} g_t^{y} = g_t^{-y_t} = -(1 + BR) \eta_t g_t$ (equation 20) which is helpful in the analysis. Sadly, it seems that there is no such relation in the multiclass setting which blocks the analysis.

---

> > ### Comment · Reviewer_TSZ4 · 2021-09-01
> > **Response**
> >
> > Thank you for the response. This helps in understanding the difficulties in obtaining a generalization of the algorithm of Jézéquel et al.

---

### Official Review · Reviewer_uuaZ · 2021-07-16

**Rating:** 7
**Confidence:** 3

**Summary:**

In this paper, the authors consider some online prediction/convex optimization problems, where it is assumed that the losses satisfy a mixability condition, as well as additional assumptions inspired by logistic regression.
While these problems can be addressed by Vovk's aggregating algorithm, the resulting procedure would be computationally expensive and involved.
Efficient and more direct procedures have been proposed for least-squares regression by Vovk and Azoury-Warmuth (VAW), and more recently for (binary) logistic regression by Mourtada and Gaiffas (MG) and Jezequel, Gaillard and Rudi (JGR).
The present paper sets out to extend these constructions; the considered losses are assumed to be mixable, and satisfy conditions such as smoothness and a (weakened) self-concordance satisfied by logistic regression (lines 215-223).
A procedure for this setting, called Gaussian aggregation forecaster (GAF), is proposed.

The main outcomes are the following:
- In Sec 4.1, for linear regression the authors recover the VAW procedure and its guarantee. [However, there seems to be a possible issue with this derivation.]
- In Sec 4.2, the authors consider multi-class logistic regression, and obtain a procedure and guarantee for this problem. These extend those of MG and JGR for binary logistic regression to the multi-class setting. They also go beyond what can be achieved by procedures such as online gradient descent, online Newton step or follow-the-regularized-leader. This is the main result of this work.

**Ethical Concerns:**

None, as far as I can tell

**Limitations And Societal Impact:**

Yes

**Main Review:**

At present, I do not have a particularly strong recommendation for this submission.

On the one hand, the paper is overall very clear, and fairly well-written (apart from some small typos). While I did not check the detailed proofs in the supplementary material, the provided sketch in the main text appears to be correct, the main arguments are the right ones (similar to prior work), and the bounds looks the way they should, so I would say that the paper seems correct.

On the other hand, there are two main possible limitations:
- It seems that as stated, the derivation for square loss may be incorrect. The reason for this is that the predictions of linear functions need not be bounded by the range Y of outputs y_t's. This can compromise the application of mixability, since the square loss is only mixable on bounded intervals (so both the outputs and prediction need to be bounded). One could in principle fix this by clipping the predictions of linear functions before aggregating, but then the resulting might not have a closed-form expression (?) and would not in any case correspond to VAW. I think this part should be fixed or adapted/removed.
- At a high level, the procedure, assumptions and analysis seem to essentially "abstract" those of JGR, extending them to general losses satisfying the conditions that are used in the analysis. The main extension from the works of MG and JGR on binary logistic regression is to handle the multi-class setting; while this is a practically relevant extension, it might perhaps also be a slightly limited one.

This being said, I would be happy to increase my score if the authors could: (1) describe a possible fix for the square loss case (or correct me if I got something wrong); and/or (2) point out to specific insights or difficulties in their setting, beyond those that appear in binary logistic regression.

---
Minor comments

- In the abstract and line 44, the provided regret bounds depend on sample size n and parameter norm B, but the dependence on dimension d is left implicit; it would perhaps be better to indicate it (e.g., O(d log(Bn)) instead of O(log(Bn))), as correctly done in other parts of the text.
- In addition to Foster et al., bounds for logistic regression through Vovk's aggregation algorithm were also obtained by Kakade and Ng "Online Bounds for Bayesian Algorithms".
- Line 112: "there exists"
- Line 125: "the most original assumptions"
- On a minor note: the optimal mixability constant for square loss is obtained by combining predictions in a non-linear way; if one uses the linear/convex combination (which typically leads to a closed-form expression), then the resulting constant is 4 times larger. This makes little difference, but it may be worth checking that the correct constant is used--this may already be the case, but please verify it.

----
Update after authors' reply:
Thank you for the clarification about square loss. This addresses my concern about the boundedness issue, which turns out to be ill-funded. I have updated my evaluation to account for this.

**Time Spent Reviewing:**

5

---

> ### Author Response · Authors · 2021-08-10
> **Response to reviewer uuaZ**
>
> We thank the reviewer for the constructive comments. We address the reviewer’s main points below and hope our argument will make the reviewer re-evaluate our paper:
>
> - We thank the reviewer for pointing out one possible source of confusion about the boundedness of $\hat{y}_t$, that we are going to clarify now. Indeed, if we used the exp-concavity of the square loss, it would have implied that both the output $y_t$ *and* the prediction $\hat{y}_t$ need to be bounded. In this paper, however, we don’t assume exp-concavity, but mixability. In the case of mixability the situation is different. In this case squared loss does still require boundedness of the observed output $y_t$, but does *not* require the boundedness of the prediction $\hat{y}_t$ (which is important because VAW does not have bounded prediction). To check this affirmation you may refer to Section 2.4 of  Vovk (2001). Since in this paper we use only mixability (and not exp-concavity), we don’t require bounded prediction for squared loss. We will take this as an occasion to explain it better also in the text in Section 4.1.
>
> - We highlight that the proposed multiclass algorithm is substantially different from the binary classification algorithm of JGR and based on different principles. Indeed we tried hard for few months to find natural extensions to such algorithm, before finding the following inherent difficulties that make such a road hardly practicable. On the intuitive side, Jézéquel et al. was based on the observation that when $\theta_t$ is far from 0 (let say $\theta_t >> 0$) either $y_t = 1$ and the curvature was advantageous or $y_t = -1$ and $\hat \theta_t$ tends to the oracle $\theta_{t+1}$. This intuition is a bit lost in multiclass setting as several oracles are possible if $y_t \neq 1$. On a more technical side, if we extend Jézéquel et al. in a natural way we pay $g_t^\top A_t^{-1} \sum_{y \neq y_t} g_t^{y}$. In the binary case $\sum_{y \neq y_t} g_t^{y} = g_t^{-y_t} = -(1 + BR) \eta_t g_t$ (equation 20) which is helpful in the analysis. Sadly, it seems that there is no such relation in the multiclass setting which blocks the analysis.
>
> - Concerning the relevance of considering the multiclass setting, we see three motivations :
>   - first, multiclass classification is very common in practice so extending previous results to this setting may increase the impact of this theoretical research.
>   - as explained above, it was not obvious how to extend previous works. It has required new ideas that could be of interest on their own.
>   - finally, this work could initiate several follow-up. Indeed, the generality of our method should make it easier to apply to other losses than previous work. Moreover, several natural extensions of multiclass logistic could be considered like the one in Foster et al. (2018) (bandits, boosting).
>
> - We thank the reviewer for pointing out the paper of Kakade and Ng “Online Bounds on Bayesian Algorithms” that we will cite in the previous works section. Moreover, we thank the reviewer for the minor comments, that we will consider in the camera ready version of the paper.

---

> > ### Comment · Reviewer_uuaZ · 2021-08-23
> > **Thanks for clarification**
> >
> > Thank you for the clarification about square loss.
> > I have updated my evaluation accordingly.

---

### Official Review · Reviewer_4Prw · 2021-07-19

**Rating:** 7
**Confidence:** 4

**Summary:**

The authors propose a new computationally efficient algorithm for multi-class logistic regression. The propose method is based on a tight quadratic approximation of the loss function (extended from Lemma 5 of Jezequel et al.) and utilizing the mixability of the logistic loss (and other properties) to compute the prediction like AA of vovk. They show that their regret bound does not have the infamous exponential constant while being computationally efficient (n^4).


**Ethical Concerns:**

no concerns.

**Limitations And Societal Impact:**

the authors did not provide it, but I doubt there is a negative societal impact from this.

**Main Review:**



Originality 4/5: using the quadratic approximation to compute AA prediction fast is novel.

Quality 4/5:

Clarity 4/5:

Significance 4/5: performing online logistic regression with statistical & computational efficiency is of great interest in theory and practice.

The paper is very clear about the contribution and it is concise. The proof of the quadratic lower bound (Lemma 4) seems much simlper than then one from Jezequel et al. The solution is novel. While $n^4$ time complexity would not belong to a practical algorithm, I believe it achieves an important mildstone.

A question: What would be the difficulty of extending Jezequel et al. for the multi-class case? We can use Lemma 4 and perform similar improper learning, right?

Another question:  Q: what's the importance of using the actual loss function for the last iterate $t-1$ when computing the estimator?

[etc]

* L40 $\Phi(x)$: did you define this before L40?
* L178: was $L_n(\theta)$ defined before?
* L178: "$L_t(\theta) \ge \tilde L_t(\theta)$ ": I think $L_t(\theta)$ here should be $L_t(\theta) + \lambda \|\theta\|_2^2$.
* L212: it would be easier to follow if the range of $i$ and $j$ is provided here.

[after rebuttal]
Thanks for explaining why Jezequel et al.'s technique is not easily transferred. I am fully satisfied with the rebuttal given by the authors.

**Time Spent Reviewing:**

5

---

> ### Author Response · Authors · 2021-08-10
> **Response to reviewer 4Prw**
>
> We thank the reviewer for the constructive comments. We are glad that the reviewer considers our work as an important milestone in the quest for computationally efficient algorithms in this field. We will answer to the posed questions in order:
> - It is true that we can naturally generalize the algorithm of Jézéquel et al. to the multiclass setting using Lemma 4. However, it is not clear if this generalized algorithm would have the same guarantee as in the binary setting. Indeed, we have encountered the following difficulties when trying to generalize the analysis. On the intuitive side, Jézéquel et al. was based on the observation that when $\theta_t$ is far from 0 (let say $\theta_t >> 0$) either $y_t = 1$ and the curvature was advantageous or $y_t = -1$ and $\hat \theta_t$ tends to the oracle $\theta_{t+1}$. This intuition is a bit lost in multiclass setting as several oracles are possible if $y_t \neq 1$. On a more technical side, if we extend Jézéquel et al. in a natural way we pay $g_t^\top A_t^{-1} \sum_{y \neq y_t} g_t^{y}$. In the binary case $\sum_{y \neq y_t} g_t^{y} = g_t^{-y_t} = -(1 + BR) \eta_t g_t$ (equation 20) which is helpful in the analysis. Sadly, it seems that there is no such relation in the multiclass setting which blocks the analysis.
> - On the importance of using the actual loss $\ell_{t-1}$ for computing $\theta_t$ : our approach is based on quadratics approximations of losses, so the question we face is where to do such approximation. Doing the approximation of $\ell_{t-1}$ at $\theta_t$ as defined in equation 2 leads to $\nabla \tilde L_{t-2}(\theta_t) + \nabla \ell_{t-1}(\theta_t) = \nabla \tilde L_{t-1}(\theta_t) = 0$ and as a consequence $\theta_t = \arg\min_\theta \tilde L_{t-1}(\theta)$ which is helpful in the analysis. One may do the approximation elsewhere (at $\theta_{t-1}$ for example), but a new term would appear in the analysis which would need to be controlled.
> - Thanks for the minor comments that we will take into account. We use the convention of including the l2 regularization into the definition of $L_t$. We will add the missing definitions.

---

### Decision · Program_Chairs · 2021-09-27

**Decision:**

Accept (Spotlight)

**Comment:**

This paper makes significant progress on the problem of computationally efficiently achieving logarithmic regret for multi-class logistic regression. The closest comparator work of Foster et al. (2008) achieves regret $O(d K \log (B n))$ but at the prohibitive $O(n^{37})$ total cost. In contrast, ignoring other factors, the present paper devises the algorithm Efficient-GAF, which obtains regret $O(d K (B^2 + (\log(K) + B) \log(n))$ at a total runtime that is $O(n^4)$, ignoring other factors. Although the regret of the present work is exponentially higher in $B$ relative to (Foster et al., 2008), the regret is also exponentially lower than the $O(e^B \log n)$ regret achieved by Online Newton Step, while still keeping the runtime reasonable. Although Efficient-GAF's $O(n^4)$ runtime is not yet practical by some standards, this is still immense progress towards the goal of a practical runtime.

This work contains several techniques that appear to be original and also could see use in future works. Also, as the authors mentioned in the discussion phase, there is a fundamental roadblock in trying to generalize the results of Jézéquel, Gaillard, and Rudi (COLT 2020), hereafter referred to as (JGR), to the multi-class setting, and the reviewers agreed with the authors' assessment here. Given that all four reviewers asked whether such a generalization is possible, I would like to suggest that the authors, either in the paper, or in an appendix, try to mention the issue with extending JGR to the multi-class setting. Future readers may wonder about this question otherwise.

One side note: For clean comparison to results in the binary logistic regression setting (such as the work of JGR), it is worth mentioning that both of the results of Foster et al. (2008) and the present paper hold with high probability and the runtime dependence is logarithmic in $1/\delta$, whereas the work of JGR gives guarantees that hold deterministically. This was not really apparent until far into the paper. Therefore, it seems that another question is whether something close to practical efficiency can be achieved deterministically in the multi-class case.

In summary, this is a strong paper that I expect to be influential, and it deserves to be accepted at NeurIPS 2021.